# The Rootstock Genotypes Determine Drought Tolerance by Regulating Aquaporin Expression at the Transcript Level and Phytohormone Balance

**DOI:** 10.3390/plants12040718

**Published:** 2023-02-06

**Authors:** David Labarga, Andreu Mairata, Miguel Puelles, Ignacio Martín, Alfonso Albacete, Enrique García-Escudero, Alicia Pou

**Affiliations:** 1Departamento de Viticultura, Instituto de Ciencias de la Vid y del Vino (Gobierno de La Rioja, Universidad de La Rioja, CSIC), Finca La Grajera, Ctra. De Burgos km 6, 26007 Logroño, Spain; 2Departamento de Nutrición Vegetal, Centro de Edafología y Biología Aplicada del Segura (CEBAS), Consejo Superior de Investigaciones Científicas (CSIC), Campus Universitario de Espinardo, Espinardo, 30100 Murcia, Spain

**Keywords:** grapevine, water stress, molecular alterations, stomatal traits and plant physiology

## Abstract

Grapevine rootstocks may supply water to the scion according to the transpiration demand, thus modulating plant responses to water deficit, but the scion variety can alter these responses, as well. The rootstock genotypes’ effect on the scion physiological response, aquaporin expression, and hormone concentrations in the xylem and the leaf was assessed under well watered (WW) and water stress (WS) conditions. Under WW, vines grafted onto 1103P and R110 rootstocks (the more vigorous and drought-tolerant) showed higher photosynthesis (A_N_), stomatal conductance (g_s_), and hydraulic conductance (Kh_plant_) compared with the less vigorous and drought-sensitive rootstock (161-49C), while under WS, there were hardly any differences between vines depending on the rootstock grafted. Besides, stomatal traits were affected by drought, which was related to g_s_, but not by the rootstock. Under WS conditions, all *VvPIP* and *VvTIP* aquaporins were up-regulated in the vines grafted onto 1103P and down-regulated in the ones grafted onto 161-49C. The 1103P capability to tolerate drought was enhanced by the up-regulation of all *VvPIP* and *VvTIP* aquaporins, lower ABA synthesis, and higher ACC/ABA ratios in leaves during WS compared with 161-49C. It was concluded that, under WW conditions, transpiration and stomatal control were rootstock-dependent. However, under WS conditions, alterations in the molecular components of water transport and hormone concentration of the scion resulted in similar gas exchange values in the studied scions grafted onto different rootstocks.

## 1. Introduction

Plants have developed physiological, cellular, and molecular responses to multiple stress conditions that enable them to adapt to these stresses. A crucial constraining factor is drought, which affects plant growth and development. Grapevine is severely hampered by drought worldwide and responds to water stress in several crucial ways. On the one hand, grapevines, as sessile organisms, may make versatile vicissitudes (such as stomatal closure or accumulation and compartmentation of osmotically active solutes) and metabolism (changes in genes expression involved in stress responses and cell metabolism) [1], which adjust the circulation of water in a rapid temporal scale. On the other hand, long-term responses are related to the growth and development of permanent phenotypic and hydraulic structures. In this case, changes in canopy transpiration and/or root water supply to moderate the water demand, and the production of xylem elements less vulnerable to embolism, are some responses that induce a high tolerance to more negative water potential gradients or limited water supplies [2].

Many authors have been encouraged to classify grapevine varieties as isohydric or anisohydric to understand the different drought tolerance detailed mechanisms between the existing genotypes [3,4,5]. Isohydric behavior has shown more stomatal sensitivity to lower soil water potential or an increased vapor pressure deficit compared with anisohydric varieties [6]. It allows the plant to maintain its leaf water potential (Ψ_leaf_) at constant ranges [7,8]. To the contrary, anhisohydric plants transpire even though soil water content decreases due to a weak stomatal adjustment capacity.

However, this classification is controversial, since multiple varieties have shown opposite performances under different climatic, edaphic, and growth conditions [9,10]. In this sense, rootstock selection is one of the variables that influence the scion characterization. Indeed, it has been shown that rootstocks modify the rate of scion growth over various time scales (reviewed by Zhang et al. [11]).

Rootstocks provide benefits in response to biotic and abiotic stresses and play a crucial role in water uptake, nutrient absorption, and drought tolerance [12,13,14,15,16]. Concerning drought tolerance, *V. berlandieri* × *V. rupestris* combinations are more tolerant (and vigorous) than *V. berlandieri* × *V. riparia* combinations. Consequently, rootstocks 1103P and R110 have been classified as drought tolerant, while 161-49C has been classified as drought susceptible [16,17,18]. Several studies performed in the same scion have reported differences between drought tolerant and drought susceptible rootstocks in terms of scion A_N_, g_s_, E, Kh_plant_, and Ψ [13,19,20,21] values, resulting in more vigorous plants. Besides, gas exchange is influenced by the stomata, specifically by their anatomical features and the pore width regulation. Genetic factors or growth conditions induce it [22,23]. It is known that varieties differ in their stomatal density and size, especially under drought conditions [24,25]. However, the effect of rootstocks on stomata is uncertain. Serra [26] found changes in stomatal size, but not stomatal density, in two drought tolerant rootstocks (R110 and R99). In addition, stomatal density and starch concentration in roots and stems are inversely correlated [27], indicating that the carbohydrate reserve status of grapevine may be an important endogenous determinant of stomatal density. Thus, it is crucial to identify the stomatal traits that trigger the best g_s_ functioning to take advantage of genotypic variation.

Moreover, other differences in hydraulic conductance between rootstocks might explain the different effects on leaf development. The root water uptake rate resulted from the combined effect of osmotic (∆π) and hydrostatic (∆P) forces, originating from the accumulation of solutes and transpiration, respectively. Water can flow along two pathways through the roots: the apoplastic and cell-to-cell paths. The cell-to-cell pathway is by simple diffusion or through aquaporins [16,28].

Several studies have shown that aquaporins act in osmoregulation and affect hydraulic conductance, influencing g_s_ [28,29,30]. Aquaporins are small membrane proteins, which are mainly water channels, but they may also transport gases (CO_2_), osmolytes, and other substances (H_2_O_2_) [31,32,33]. They belong to the major intrinsic protein (MIP) superfamily. There are 33 MIP (four incompletes) sequences identified in grapevine [34,35,36,37]. MIPs are divided into several families. The plasma membrane intrinsic proteins (PIPs) and the tonoplast intrinsic proteins (TIPs) are the most studied [34]. Many authors showed changes in aquaporin expression between varieties (Grenache, Chardonnay, and Syrah) and also between roots and leaves under moderate water stress [38,39,40]. Furthermore, Gambetta et al. [15] demonstrated the rootstock effect in root aquaporin expression under water stress. Nevertheless, the expression of aquaporins varies depending on the organ studied and the type of drought [41,42]. In this sense, more studies are needed to elucidate the response of aquaporins to severe and extended drought periods in the field [28] and their relation with root water uptake.

In addition, biochemical signals also play an essential role in the plant water deficit response and may differ between rootstocks. The root system is a source of many plant hormones [43,44]. Phytohormones regulate numerous plant processes, such as responses against abiotic stresses [45]. Abscisic acid (ABA) is the most important and studied phytohormone in plant water stress responses, and its accumulation triggers stomatal closure [4,40,46,47]. Besides, it is also involved in growth regulation and osmoregulation, preventing water loss [37,48]. The effects of ABA in grapevine under water stress are known. Despite that, the implication of other phytohormones related to growth has also recently been associated with the response to abiotic stresses in this species, including water stress [49,50]. However, their function remains unclear due to the complex action mechanism of phytohormones (crosstalk). Some of these phytohormones include indole-3-acetic acid (IAA), cytokinins (CK), gibberellins (GA), ethylene (ET), salicylic acid (SA), and jasmonic acid (JA). Conflicting results have been published on this topic. Haider et al. [49] found a concentration increase in all study hormones under water stress in a complete RNA-seq analysis. Chrysargyris et al. [51] discovered different phytohormone responses to water stress between drought tolerant (Xynisteri) and sensible (Chardonnay) varieties. Recently, Prinsi et al. [50] performed a proteomic study, showing differences in phytohormone response to drought between rootstocks M4 and 101-14. Thus, further investigations are needed to elucidate the phytohormone response to water stress in grapevine and its effect on plant physiology and molecular responses.

The present study aimed to analyze whether different rootstocks (vigorous and drought tolerant vs. non-vigorous and drought sensible) modified scion water deficit tolerance through alterations in physiology, leaf aquaporin expression, and leaf and xylem phytohormone concentration. We performed a deep analysis by measuring scion stomatal traits, leaf and plant physiology, aquaporin expression, and leaf and xylem sap phytohormone concentrations. We hypothesize that (i) the physiological response to drought varies according to the rootstock, (ii) the differential response of each rootstock to water stress is associated with different aquaporin expression patterns and phytohormone concentrations, and (iii) phytohormones and aquaporins act together to regulate the physiological responses to water stress.

## 2. Results

Water stressed (WS) plants were not irrigated during the entire experiment. All sampling and data collection were performed during veraison (D.O.Y. 188) when a severe WS was reached (between 0.05 and 0.15 mol H_2_O m^−2^ s^−1^).

### 2.1. Water Stress and Rootstock Effects on the Plant Water Status and Physiological Responses

Different physiological responses were obtained within the cv. Tempranillo plants were grafted onto different rootstocks under well-water (WW) and WS conditions (Figure 1).

Pre-dawn and midday water potentials (Ψ_PD_ and Ψ_MD_, respectively) responded similarly to water withholding, indicating strong water reductions in WS plants compared with WW plants. Moreover, plants grafted onto 1103P rootstock obtained the highest Ψ_MD_ reductions compared with R110 and 161-49C varieties (Figure 1A,B).

Accordingly, regardless of the rootstock, net photosynthesis (A_N_) and stomatal conductance (g_s_) rates showed significant reductions under WS conditions in comparison with WW plants (Figure 1C,D). Under WW conditions, both parameters showed higher values for vines grafted onto 1103P rootstock than vines grafted onto R-110 and 161-49C, which exhibited intermediate and lower values, respectively. Under WS conditions, A_N_ and g_s_ values were lower for vines grafted onto the 1103-P and 161-49C rootstocks than on the R110, therefore obtaining higher reductions in these parameters for the plants grafted onto 1103P as a consequence of water stress compared to the other studied rootstocks.

Intrinsic water use efficiency (WUEi) increased with increasing water stress. In this case, there were no differences in WS conditions, due to a similar behavior of the parameters A_N_ and g_s_ in the three rootstocks studied. The lower g_s_ values in the vines grafted onto 161-49C resulted in an increased WUEi under WW conditions compared with the other rootstocks (Figure 1E).

Water stress also reduced plant hydraulic conductance (Kh_plant_) in all rootstocks. In this case, Kh_plant_ varied between rootstocks under WW conditions, but not under WS conditions (Figure 1F), and it showed higher values in 1103P and lower in 161-49C, while R110 showed intermediate values.

### 2.2. Both Treatment and Rootstock Contribute to Stomatal Changes

Results collected over the experiment for stomatal density, width, length, and area were subjected to two-way ANOVA. On the one hand, drought stress significantly decreased stomatal length and area, but neither the width nor the density were affected by the lack of water (Table 1).

### 2.3. Water Stress and Rootstock-Induced Changes in Vitis Vinifera Gene Aquaporin Expression

The relative expressions of *VvPIP1;1*, *VvPIP1;2*, *VvPIP2;1*, *VvPIP2;2*, *VvPIP2;3*, *VvTIP1;1,* and *VvTIP2;1* were examined by semi-quantitative real-time PCR in grapevine leaves grown under WW and WS conditions for the different grafted Tempranillo vines (Figure 2).

All seven aquaporin genes showed different response patterns to water stress, depending on the rootstock (Figure 2). In this sense, the relative expression of *VvPIP1;1* and *VvPIP2;2* increased in response to water deficit in all the rootstocks. *VvTIP1;1* was up-regulated in the rootstocks 1103P and R-110 and down-regulated in the rootstock 161-49C. *VvPIP1;2*, *VvPIP2;1*, *VvPIP2;3,* and *VvTIP2;1* only were up-regulated in the rootstock 1103P, but they were down-regulated in other situations.

Moreover, under WS condition, *VvPIP1;1*, *VvPIP1;2,* and *VvPIP2;1* were highly expressed in the rootstock 1103P, while *VvPIP2;2* was more expressed both under WW and WS conditions in 161-49C than in the other rootstocks (Appendix A).

### 2.4. Water and Rootstock-Mediated Changes in Leaf and Xylem Hormonal Status

The content of seven phytohormones (ethylene precursor 1-aminocyclopropane-1-carboxylic acid (ACC), total cytokinins (CK), total gibberellins (GA), indole-3-acetic acid (IAA), abscisic acid (ABA), jasmonic acid (JA), and salicylic acid (SA)) in the scion leaves and the xylem sap of WW and WS grafted plants were measured for each rootstock combination (Figure 3). IAA only was detected in leaf samples. Therefore, no xylem IAA graph is represented in this figure. 

All the analyzed phytohormones resulted higher in leaves than in the xylem sap in both conditions (WW and WS), except for JA, which showed the opposite trend (Figure 3). Under WS, the phytohormone contents in leaves were strongly affected. ACC and ABA increased (Figure 3a,b—A and B), while CK, GA, IAA, and SA decreased (Figure 3a,b—C, D, F and G) when responding to drought. Only JA concentration remained stable for the two irrigation conditions in leaves, although it increased sharply in WS plants in xylem sap (Figure 3a,b—E). Remarkably, JA concentration in leaves was very low (between 0 and 0.5 ng × g^−1^). In leaves, ACC and ABA (Figure 3a—A,B), and, consequently, the ACC/ABA ratio (Figure 3a—H), also differed between rootstocks responding to drought. Specifically, ACC increased on the 1103P rootstock, and ABA increased in 161-49C in response to drought compared with the other rootstocks. Moreover, the ACC/ABA ratio increased in the 1103P rootstock responding to drought, while it was not affected in any of the other rootstocks.

In the xylem sap, water stress also affected the phytohormone concentrations (Figure 3b). In this case, all the analyzed phytohormones, except CK (decreased under WS conditions) and GA (remained invariable), increased when water was scarce. In the same way as JA in leaves, xylem GA concentration was low (between 0 and 0.020 ng × g^−1^).

In this case, the rootstock effect was less strong than the one obtained in the leaves. In the xylem sap, under WS conditions, 161-49C showed higher and lower ACC concentrations compared to R110 (Figure 3b—A and D). Moreover, R110 showed lower ABA concentration under WS conditions compared with 1103P.

### 2.5. Physiological Parameters That Apparently Involve Chemical Influences

There are some significate (*p* < 0.05) correlations between physiological parameters in leaves and the existence of a controlling chemical influence. In this case, ABA-g_s_ and ABA-Kh_plant_ showed negative exponential correlations (Figure 4D,F). In both cases, increases in ABA triggered a decrease in g_s_ and Kh_plant_, thus keeping the stomata closed. Besides, the IAA also had a quadratic correlation with Kh_plant_, but it was positive (Figure 4E).

Moreover, CK and GA appeared to be correlated with A_N_ increments, showing a positive quadratic relation (Figure 4A,B). Conversely, SA appeared to be directly related (quadratic) to A_N_ (Figure 4C).

## 3. Discussions

Grapevine rootstocks with contrasting vigor induced differences in growth capacity and drought tolerance to the scion adjusting the water supply to shoot transpiration demand [6,12,52]. However, the interaction and communication between rootstock and scion are still poorly understood and, yet, they are critical in improving management strategies and grafting technologies [53,54].

To explore the potential rootstock genotype’s contribution to the scion adaptive responses to water limiting conditions, scion stomatal traits, leaf water status, and its changes regarding physiology, gene expression, and hormonal signaling were evaluated.

### 3.1. Stomatal Traits Partially Explain Variations in g_s_ between Well-Watered and Water-Stressed Plants but Not between Rootstocks

Stomata are essential for photosynthesis and environmental changes response. Thus, they are tightly regulated to control leaf water loss and carbon dioxide intake. Stomatal size and density affect the functional efficiency of stomata and are strictly controlled by developmental and environmental cues to ensure a precise stomatal adjustment [55,56]. For instance, grapevine leaves stomatal density was reduced when responding to high soil temperature and atmospheric CO_2_ [27]. Moreover, in grapevine, experimental data showed that stomatal development is under genetic control with differences between grapevine cultivars [57].

However, the rootstock effects on stomatal density remain unclear. Due to the rootstocks’ influence on vigor, plant water status and leaf gas exchange, it is postulated that rootstocks may enhance drought tolerance in grapevine, affecting leaf stomatal size and density. Recently, Serra et al. (2017) studied the drought effect on stomatal density in Pinotage cv. grafted onto R110 and R99 rootstocks. They did not find differences in stomatal density between grafted plants, although a decrease in the diameter of the stomatal pore between the irrigation and the drought-treated plants was observed. Likewise, in this study, although the stomata density did not vary responding to drought, a wide variation in length and total area could explain the maximum g_s_ in WW plants compared with WS plants (Table 1, Figure 1). By contrast, although the rootstocks 1103P and R110 promoted higher growth and vigor than 161-49C [58], no genetic variation in stomatal density, width, or length differences were observed (Table 1). Hence, the data indicate that the stomatal traits of grafted plants in the different rootstocks and their maximum g_s_ deviate considerably, inadequately reflecting the operating pore area. In this case, stomatal regulation could be achieved by opening and closing the stomatal pore, thereby either increasing or reducing stomatal conductance or the rate by which water or CO_2_ is exchanged [59], instead of by modulating the frequency at which stomata develop in new organs.

### 3.2. Grapevine Physiology Changes Depending on the Rootstocks and Treatment

There are different rootstock classifications, some based on their vigor and others on their drought resistance ability [16,26,60]. In this sense, rootstocks 1103P and R110 have been reported as drought-tolerant and vigorous, while 161-49C is reported to be drought-sensible with moderate vigor [13,19,61,62,63,64]. The physiological response of both groups has been investigated previously, showing that more vigorous rootstocks trigger lower leaf water potential (Ψ) and higher net photosynthesis (A_N_), stomatal conductance (g_s_), transpiration (E), and plant hydraulic conductance (Kh_plant_) than the less vigor ones [21,62,63,65,66]. Accordingly, the obtained results supported that the scions grafted onto the rootstocks 1103P and R110 had higher A_N_, g_s_, and Kh_plant_ under irrigation (Figure 1C,D,F), which promotes higher growth and vigor of these plants (Tabla S1). The explanation could be either by the rootstock capacity for uptaking water or by the cavitation vulnerability, accompanied by different gradients in xylem tracheid diameters depending on its vigor and deep. Indeed, Jackson et al. [67], for *Juniperus ashei,* reported more than four times greater mean conduit diameters in deep roots than in shallow roots, highly contributing the deeper roots to the whole plant water use. This study emphasizes the root hydraulic architecture importance for water uptake at a range of depths.

Contrarily, the three studied rootstocks responded similarly and obtained the closest values under drought (Figure 1). However, the higher the vigor, the higher the drop in Kh_plant_. This suggests that some cavitation, if any, was present under the obtained WS conditions in the vigorous rootstocks. In this regard, Lovisolo et al. [4] described a cavitation threshold of −1.5 MPa in grapevine and, in this case, we obtained leaf water potentials of approximately −1.7 MPa. Accordingly, a similar stomata ability to control water loss was observed in all three rootstocks, following the same trend as the one for Kh_plant_. This fact may be partially explained by the limited stomatal dimensions (either in length or in the stomatal area) in WS plants, which, in turn, were the same in all the rootstocks, or because other chemical or gene-expression patterns permit to regulate their water status against desiccation.

Similarly, there were differences in intrinsic water use efficiency (WUEi) between rootstocks only in WW conditions. In this case, rootstocks 1103P and R110 showed lower WUE_i_ than 161-49C (Figure 1E). Similar results were also reported by Romero et al. [19] using the cv. Monastrell grafted onto 1103P, R110, 41B, 140Ru, and 161-49C, suggesting that the more vigorous rootstocks, with larger canopy (Tabla S1), need more water and consume their water reserves faster than the less vigorous ones.

Altogether, these results suggest a rootstock influence on the scion behavior, especially under well-watered conditions. In this sense, vigorous rootstocks would be encouraged to keep their stomata more open, thus maximizing the photosynthetic rate, while the less vigorous ones would do the opposite.

When water is scarce, other responses may be crucial to maintain plant homeostasis. We suggest that leaf aquaporin expression and hormonal status respond to water stress, resulting in similar leaf water potential, gas exchange, and Kh_plant_ in all grafted plants.

### 3.3. Aquaporin Up-Regulation Promotes Tolerance to Water Stress

The root system controls the water uptake and transport to the leaves, as well as being involved in scion stress sensing and signaling. Hence, it is considered to play a central role in the modulation of water stress responses [68]. Thus, root hydraulics may have a significant role in the regulation of transpiration, where root water channels (aquaporins; AQPs) may play an important role [28,29,38,40,69]. However, although much research has been conducted, it is still unclear how AQPs affect transpiration. Most studies demonstrated an AQP correlative effect on leaf gas exchange triggered by root AQPs [70]. However, not only have we evaluated the possible regulation of root AQPs by leaf transpiration via hydraulic signals, but we have also evaluated the direct leaf AQP involvement in plant homeostasis. In this sense, some studies reported different AQP expression patterns between leaves and roots [71,72]. Generally, under drought conditions, AQP expression patterns are reported to be higher in roots than in leaves, although both are crucial [28,42,73,74]. Besides, reduced and increased AQP expression against WS has been shown in leaves and roots, respectively [4,15,30,35,38,39,41,72,75,76]. Gambetta et al. [47] showed an increased expression of root AQPs in a rootstock study, which was directly related to root hydraulic conductivity (Lp_root_).

The present work showed how, under WW conditions, 1103P showed a greater capacity to sustain lower leaf potential (Ψ_leaf_) and higher A_N_, g_s_, and Kh_plant_, while limiting the decrease in these parameters during drought compared with the other graft combinations, since all of them achieved the same values.

In this sense, although the reduction percentage of WS plants compared to WW ones was higher in this rootstock, the final value obtained was the same for all rootstocks. At the same time, we observed that, during WS, *VvPIP1;1*, *VvPIP1;2,* and *VvPIP2;1* were more abundantly expressed in 1103P than on the other grafted plants (Appendix A), and all the evaluated leaf AQPs were up-regulated in 1103P. Nevertheless, almost all of them (except *VvPIP1;1* and *VvPIP2;2*) were down-regulated in 161-49C (Figure 2). 

Aquaporin up-regulation under severe and prolonged water stresses, such as the one in this study, has also been previously reported in Galmés et al. [72]. In this study, the authors showed an aquaporin up-regulation under severe stress to encourage water uptake due to its extreme lack. Similarly, we may argue that, during water stress, Tempranillo plants responded with a tight regulation of leaf AQPs to compensate for the lack of water when grafted onto 1103P due to increased water demand (due, in turn, to the higher canopies). Zarrouk et al. [41] also showed up-regulation of *VvPIP2;1* in leaves and its involvement in osmoregulation. For instance, the expression of *Vitis* APQs in the aerial part increased under severe water stress to limit water loss and keep water content constant for all grafted plants.

Research comparing varieties has shown that near-isohydric varieties down-regulate aquaporins, while near-anisohydric varieties up-regulate them [38,39,47]. Assuming a rootstock 161-49C near-isohydric behavior and 1103P and R110 near-anisohydric behavior under these conditions of WS, the results are similar to those of the previous authors.

### 3.4. Hormones Alteration and Interaction

#### 3.4.1. ABA and ACC Increase in the Same Way Responding to Drought

The rootstock-induced drought tolerance in grapevine also includes phytohormonal-mediated responses. Several works confirmed that plant growth regulators are involved in drought response regulation. For instance, the principal plant drought response is stomatal closure, which may be abscisic acid (ABA)-dependent or ABA-independent [46]. Also, Parent et al. [77] suggest that ABA has long-lasting effects on plant hydraulic properties via aquaporin activity, which contributes to maintaining a favorable plant water status.

Our data suggest a clear relationship between ABA and both g_s_ and Kh_plant_, where an increase in ABA triggers a decrease in g_s_ and Kh_plant_ until reaching a plateau (Figure 4D,F). Under WW conditions, either ABA and ACC concentrations, both related to drought resistance [47,78,79,80,81], kept similar concentration for all the studied rootstocks, both in the leaf and in the xylem. However, under WS conditions, increases in both phytohormones were rootstock-dependent. The plants grafted onto rootstock 161-49C showed higher leaf ABA concentrations compared to 1103P and R110 (Figure 3a—B), while plants grafted onto the rootstock 1103P showed higher leaf ACC (Figure 3a—A). On the one hand, drought tolerant rootstocks [21,68], such as 1103P, have a lower capacity for leaf ABA accumulation, which may explain the obtained results, although the xylem ABA concentration recorded the greatest levels, allowing to keep its stomata closed at a similar level as in 161-49C.

On the other hand, ACC is an ethylene precursor, a molecule involved in growth inhibition, root elongation, and leaf abscission induction [82,83,84,85]. Therefore, higher concentrations of this phytohormone in the scion grafted onto the most-vigorous rootstock (1103P) may promote lower growth under WS.

Accordingly, the ACC/ABA index showed higher levels in the scion grafted onto the rootstock 1103P. Albacete et al. [86] pointed out that the ACC/ABA index is an indicator of leaf senescence and deterioration since both, in one way or another, decrease plant activity and induce senescence and ripening processes. Therefore, the obtained differences in this index were rootstock-dependent, which indicates that, in this case, part of the scion drought tolerance was conferred by the rootstock.

#### 3.4.2. Photosynthesis Is Influenced by GA and CK

Several studies have shown that GA and CK are involved in plant photosynthesis regulation and source/sink balance [79,81,87,88,89]. Concretely, the movement of CK in the transpiration stream from the roots to the shoots stimulates the expression of photosynthesis genes [90]. 

However, disparate results have been found regarding the effect of drought on GA and CK concentrations. Iqbal et al. [91] and Salvi et al. [78] reported a decrease in GA and CK under water stress in several species, although some discrepancies exist depending on the species and the stress conditions regarding CK concentrations [92]. In grapevine, Haider et al. [49] found increased GA concentration during drought. Contrary, Prinsi et al. [50], and Giacomelli et al. [93,94], showed a decrease in GA under drought. Accordingly, we have shown a clear relationship between the obtained A_N_ rates and the GA and CK concentrations (Figure 4C,D). In both cases, reductions in A_N_ during the imposed drought imply a decrease in the concentration of these phytohormones. Moreover, in this study, reduced plant growth (pruning and berry weight in Appendix A) conditions may also be associated with a decrease in GA and CK concentrations under WS, as previously reported [78,81]. These authors showed how low CK concentrations in WS induced leaf and fruit abscission, which prevent water loss through the leaves.

In this case, contrary to the ABA and ACC responses, the rootstock seems to influence the GA and CK concentration under irrigation, but not under drought. Under irrigation, the rootstock 1103P obtained the lowest concentrations in the leaf of both phytohormones, while R110 and 161-49C had the highest (Figure 3a—C,D). On the other hand, it does not agree with the higher A_N_ rates obtained for the 1103P rootstocks under WW conditions. Moreover, in grapevine, Lucini et al. [95] showed higher GA and CK concentrations in a drought-resistant rootstock, which improved the plant’s response to WS. Boonman et al. [96] found that more shaded leaves had lower CK concentrations in tomatoes. It could explain the results of rootstock 1103P, which had a higher canopy and, hence, greater shading. Indeed, it needs to be further studied because, as in this study, reduced levels of GA and CK in the most vigor rootstock (1103P) are shown under WW conditions. This could be explained by the fact that GA and CK growth signal is maintained over time in this rootstock rather than being a peak signal.

#### 3.4.3. JA and SA Are Also Crucial in Abiotic Stress Response

Some of the most relevant phytohormones mediating biotic stress responses are salicylic acid (SA) and jasmonic acid (JA). SA accumulates before infection and is associated with biotrophic factors, while JA accumulation is after infection and is related to necrotrophic pathogens [97,98]. Moreover, their role in abiotic stress response in plants has been suggested [79,81]. In this study, the WS treatment resulted in a null on the leaf JA concentration or an increase in its xylem concentration. Accordingly, other studies also suggest an increase in JA under WS in several plant species [45,78,91,99], including grapevine [49,50]. JA has been implicated, together with ABA, in stomatal regulation in WS. However, in this study, only the JA xylem concentration seems to coincide with this statement, but this is not the case for the leaf. It suggests that JA is not the main regulator of the stomatal opening under WS, and/or its effect is delayed compared to that of ABA.

Regarding the SA response under WS conditions, we have found some discrepancies between the leaf and the xylem. Leaf SA decreases were found responding to WS, but the opposite was obtained in the xylem. In spite of this, it has been previously reported that phytohormone concentrations in leaf extracts were positively and significantly correlated with those of xylem sap in grafted tomato plants [100], and the inverse correlation was observed in this study for SA, which could indicate that this phytohormone first accumulates in the xylem and is then transported to the leaf. It may explain the contrasting results observed in the literature for the different species, such as chickpeas, tomatoes, *Arabidopsis*, or *Brassica* sp., concerning SA response to WS [78,79]. Some studies have shown that high SA concentrations affect A_N_ enhancement, antioxidant production, and stomatal regulation. Indeed, we obtained a good correlation between leaf SA and A_N_, in which higher SA concentrations resulted in increased A_N_ (Figure 4C)_,_ although, neither for JA nor SA, did the rootstock seem to be a determining factor. In this case, the phytohormonal response was the same, regardless of the rootstock.

#### 3.4.4. Leaf Auxins Role in Drought Resistance

Auxins, particularly indole-3-acetic acid (IAA), are phytohormones related to the elongation and cell division of the root system and the rest of the plant [84,101]. IAA concentrations only were detected in leaves, and the values found were very low (between 0 and 0.12 ng g^−1^). 

Drought caused an IAA decrease in the leaf in all the studied rootstocks (Figure 2a—G). However, disparate results have been observed for IAA responses to drought. Hussain et al. [99] showed a negative IAA response in *Arabidopsis thaliana*, while Ahmad et al. [79] reported IAA increases in *Ozyria sativa*. In grapevine, different studies also reported different IAA responses induced by WS [50,82,102]. An IAA decrease would result in reduced root and plant growth, thus avoiding water loss. Furthermore, reactive oxygen species (ROS), released under stress conditions, such as drought, trigger ABA increase and IAA decrease (directly or indirectly), which support our results [99]. This way, we observed a negative correlation between ABA and IAA. We also noticed a positive relationship between IAA concentration and Kh_plant_ (Figure 4E). Contrarily, Vandeleur et al. [76] did not observe a correlation between IAA and Lp_root_. Other studies [46,103] showed a negative correlation between this phytohormone and root and cell hydraulic conductivity. Nevertheless, auxin has been demonstrated to increase the water permeability of leaf epidermal cells of *Allium cepa* bulbs and *Rhoeo discolor* [104].

Under irrigation, rootstocks R110 and 1103P registered the highest and lowest concentrations, respectively, while no differences were found in WS (Figure 2a—G). Higher IAA concentrations of more resistant grapevine rootstocks in grapes are reported to trigger higher development and earlier ripening [105], which in turn does not correspond with the obtained results, in which the most vigorous rootstocks obtained lower IAA concentrations. This could indicate a lower IAA concentration, but which is maintained over time, or this could indicate that the IAA peak occurred before the harvest of the samples.

In conclusion, we show that the resistance capacity of vines provided by rootstocks under severe and prolonged stress is a consequence of alterations at the molecular level more than the physiological level. Although the physiological parameters of the vines grafted onto the different rootstocks differed under well water conditions, they were similar under water stress. Under water stress, vines grafted on 1103P kept their drought tolerant status through aquaporins overexpression (*VvPIP* and *VvTIP*) and changes in phytohormone concentrations (ABA showed the lowest values under stress), while the opposite happened in vines grafted on 161-49C, which was associated with their drought susceptibility. In summary, this study provides interesting results that allow rootstock characterization against drought and facilitates their selection according to the geographic area and climate. Besides, it highlights the importance of molecular alterations in the face of severe droughts, such as those predicted for the coming years. This fact may help to target future research related to drought in the grapevine. On the one hand, it indicates that it is essential to study the scion aquaporins apart from the root ones, as they play a decisive role in water stress. Not only *VvPIP* and *VvTIP* should be taken into account, but so should other families of MIP that have been less studied so far. On the other hand, this shows the importance of several phytohormones in response to drought in addition to the well-known and studied ABA. In this sense, phytohormones, such as ACC, GA, CK, IAA, and some not always linked to abiotic stresses, such as SA and JA, aslo appear relevant in the response to drought response. However, only some of them, such as ACC and GA, CK, and IAA (only in WW) seem to respond in a dependent manner to the rootstock. Rootstock growers’ selection against pathogens and pests is widely known. Nevertheless, its implementation to face abiotic stresses, such as drought, is not so common. It should be studied deeply and taken into account in the future as a natural tool against drought, one of the main effects of climate change.

## 4. Materials and Methods

### 4.1. Experimental Design

The study was performed in 2021 in a 30 year-old commercial vineyard located in Aldeanueva de Ebro, La Rioja, Spain. It is a semiarid winegrowing region of D.O.Ca Rioja, where the climate is between warm and temperate, and drought periods usually last from May to September. The mean annual rainfall is about 500 mm per year. There is loam soil, according USDA classification, composed of 23.1% clay, 45.2% silt, and 31.7% sand. Besides, there is 0.94% of organic matter and a pH of 8.3. The training system was a vertical shoot positioning (VSP), and the vines were spur-pruned on a bilateral Royat Cordon system, leaving an average of six spurs (ten-twelve buds) per plant. The vineyard spacing was 2.7 m between rows and 1.3 m between vines.

*Vitis vinifera* cv. Tempranillo was grafted onto three different rootstocks: 1103P, R-110, and 161-49C. Characteristics of the rootstocks selected differed since there are two vigorous and drought tolerant (1103-P and R-110) and one least vigorous and drought susceptible (161-49C). The formers are composed of a cross between the *V. rupestris* × *V. berlandieri* genotypes, while the latter is composed of *V. riparia* × *V. berlandieri*. The rootstocks selected were randomly distributed (48 vines per replicate) in two plots. In addition, the study rows were separated by another row composed of *Vitis vinifera* cv. Tempranillo grafted onto R-99. The two plots were differentially treated: plot 1 (well-water (WW)) had an irrigation dosage adjusted by a drip system up to 40% of the potential evapotranspiration (RTP); plot 2 (water stress (WS)) was not irrigated at all during the entire experimental period, which lasted from April to May.

All sampling and measurements were performed when there was severe water stress (0.05 mol H_2_O m^−2^ s^−1^ < g_s_ < 0.15 mol H_2_O m^−2^ s^−1^) under drought treatment (D.O.Y. 190) [106].

### 4.2. Leaf Gas Exchange and Grapevine Water Status

Physiological parameters were measured in six plants of each rootstock and treatment. Grapevine water status was followed by measuring predawn and midday leaf water potential (Ψ_PD_ and Ψ_MD_, respectively) using a Scholander pressure chamber (Soilmoisture Equipment Corp., Santa Barbara, CA, USA). Ψ_PD_ was interpreted as a proxy for Ψ_soil_, and Ψ_MD_ was taken as Ψ_leaf_. On the same day, between 11:00 a.m. and12:00 a.m., stomatal conductance (g_s_), transpiration rate (E), and net photosynthesis (A_N_) were measured on mature, healthy, and sun-exposed leaves from the same six plants using a portable open gas exchange system (Li-6400; Li-Cor Inc., Lincoln, NE, USA). CO_2_ concentration in the cuvette was set to 400 mmol CO_2_ mol^−1^ air. Intrinsic water use efficiency (WUE_i_, A_N_/g_s_) and whole plant hydraulic conductivity (Kh_plant,_ E/(Ψ_soil_ − Ψ_leaf_)) were calculated from gas exchanges and water potential values considering Ohm’s law analogy for the soil–plant–atmosphere continuum [107].

### 4.3. Stomatal Density and Aperture

Stomatal traits were studied in the same six plants, where gas exchange and grapevine water status were measured. Measurement data were collected from one sun-exposed leaf per plant, located in the main shoot between the nodes 8 and 9. A transparent nail polish peel print was made of approximately 1 × 1 cm area to the abaxial side each leaf, avoiding the mid-vein and then removed after it had dried for 1 min. All prints were taken off with the help of clear packing tape. The nail-polish imprints were mounted on microscope slides. A microscope (Nikon Ni-E) equipped with a 40× objective and a Nikon camera (Nikon DS-Ri2) was used to observe and capture the imprints for stomatal density and apertures quantification. Stomatal aperture, density, width, length, and area were measured by using NIS-Elements Br software. In this case, three 40,000 µm^2^ areas were analyzed from each slide. 

### 4.4. Leaves RNA Extraction and cDNA Synthesis

Leaves of similar developmental stage and from similar position on those used to determine the plant water status and gas exchange were harvested between 10:00 a.m. and 11:00 a.m. Four replicates per rootstock and treatment were sampled from eight different plants. Each replica was the result of the pool of two plants. The first six plants from which replicates 1, 2 and 3 were formed were the same in which gas exchange and grapevine water status were measured, while replicate 4 was obtained from the leaves of 2 randomly selected plants in the experimental row. Special frozen clamps were used to collect a total of three 1.5 cm radius disks from three different leaves per plant and stored directly at −80 °C until RNA extraction.

The disks were separately grinded in a mortar with liquid nitrogen. Total RNA was extracted from 100 mg grinded leaf tissue using Spectrum Plant Total RNA Kit (Sigma-Aldrich, St. Louis, MO, USA), avoiding DNA contaminations by on column DNase treatment (TURBO™ DNase, Invitrogen, Carlsbad, CA, USA) during RNA extraction according to the manufacturer’s recommendations. RNA quality and quantity were measured using Nanodrop Spectrophotometer ND-1000 (NanoDrop products, Wilmington, DE, USA) and the integrity of the RNA by visual inspection of rRNA banding, following 2% agarose gel electrophoresis. First-strand cDNA was synthesized by reverse transcription from 1 µg/µL of RNA using iScript^TM^ cDNA synthesis kit (Bio-Rad, Hercules, CA, USA) following the manufacturer’s instructions: 25 °C for 5 min, 46 °C for 20 min, 95 °C for 1 min, and hold at 4 °C.

### 4.5. Aquaporin Expression

Quantitative real-time PCR was performed in a 20 µL mixture containing 5 µL of diluted (1:100) cDNA, 10 µL of Applied Biosystems PowerUp SYBR Green Master Mix (Thermo Fisher Scientific, Vilnius, Lithuania), 1 µL of each primer (10 µM), and 3 µL of DEPC water in the Applied Biosystems™ 7500 Fast Dx Real-Time PCR using the following cycle: one cycle of 2″ at 50 °C and 10″ at 95 °C followed by 40 cycles of 15″ at 95 °C and 1′ at 60 °C. Before melt curve analysis, a final denaturation step of 15″ at 95 °C was performed. The melt curve was between 60 and 95 °C at 1% increments. 

Applied Biosystems™ 7500 Fast Dx Real-Time PCR software calculated the mean fluorescence threshold value (Ct) from four independent biological replicates, each with three RT-PCR replicates. Gene expression was calculated relative to the mean of the ubiquitin, elongation factor, and actin reference genes using the 2^−∆∆Ct^ method [108]. Briefly, for each rootstock, we obtained four technical and two biological replicates. For each gene, the means between biological replicates were calculated for each technical replicate. Then, the ∆Ct was calculated as the difference between the mean Ct of each technical replicate and the mean Ct of the three housekeeping. The ∆∆Ct was then calculated as the difference between the mean ∆Ct of the technical replicates of the control gene (irrigation) and the mean ∆Ct of the technical replicates of the treated gene (drought). Finally, the 2^−∆∆Ct^ was determined. The primer sequences used are shown in Appendix A.

### 4.6. Hormone Analyses

Hormone levels were analyzed in both xylem and leaf in four of the six plants where gas exchange and grapevine water status were measured. 

Xylem was sampled from the same leaves used to measure Ψ_MD_. Leaves were exposed to high pressure with a Scholander pressure chamber (Soilmoisture Equipment Corp., Santa Barbara, CA, USA) to collect the xylem with a micropipette in a 1.5 mL tube. The collected xylem was immediately frozen with liquid nitrogen and stored at −80 °C until analysis.

Leaves were sampled with special frozen clamps, which allow collecting and immediately froze 1.5 cm radius leaf disks. Three disks per plant, one per leaf, were collected and immediately kept in liquid nitrogen. In the laboratory, leaf disks were separately grinded in a mortar with liquid nitrogen, and 100 mg were stored in 2 mL tubes at −80 °C until hormone analysis.

Active cytokinins (trans-zeatin, TZ, zeatin riboside, ZR and isopentenyl adenine, iP), gibberellins (GA1, GA3 and GA4), indole-3-acetic acid (IAA), abscisic acid (ABA), salicylic acid (SA), jasmonic acid (JA), and the ethylene precursor 1-aminocyclopropane-1-carboxylic acid (ACC) were analysed according to Albacete et al. [86], with some modifications. Briefly, 1 mL of extraction solution (cold (−20 °C) methanol/water (80/20, *v*/*v*)) was mixed with 100 mg of plant material and centrifuged (20,000× *g*, 15 min) to separate solids. Then, 1 mL of the same solution was added, and the solids were re-extracted (30 min at 4 °C). Lipids and part of vegetal pigments were removed by passing the supernatants through Sep-Pak Plus †C18 cartridge (SepPak Plus, Waters, Mildford, MA, USA). Next, they were evaporated at 40 °C under vacuum. The residue was dissolved using an ultrasonic bath in 0.5 mL of extraction solution. The dissolved samples were filtered through 13 mm diameter Millex filters with 0.22 μm pore size nylon membrane (Millipore, Bedford, MA, USA).

10 μL of the resulted extract were injected in a U-HPLC-MS system consisting of an Accela Series U-HPLC (ThermoFisher Scientific, Waltham, MA, USA) coupled to an Exactive mass spectrometer (ThermoFisher Scientific, Waltham, MA, USA) using a heated electrospray ionization (HESI) interface. Xcalibur 2.2 (ThermoFisher Scientific, Waltham, MA, USA, EE.UU.) was the software used to obtain the mass spectra. Phytohormones were quantified by creating calibration curves (1, 10, 50, and 100 μg·L^−1^), which were corrected with 10 μg·L^−1^ deuterated internal standards. Recovery percentages ranged between 92 and 95%.

### 4.7. Statistical Analyses

Two-way analysis of variance (ANOVA) was used to find out statistical differences between means. When the interaction between factors was significant (*p* < 0.05), one-way ANOVA was performed. Multiple comparisons of means post-hoc Duncan (*p* < 0.05) were carried out when there were statistical differences with SPSS 22.0 (IBM Corp., Armonk, NY, USA). Regression analyses and statistical parameters (R, R2 and *p* values) were obtained with the SigmaPlot 12.0 software package for Windows. Dates were represented with SigmaPlot 12.0 and GraphPad 8 software.

## Figures and Tables

**Figure 1 plants-12-00718-f001:**
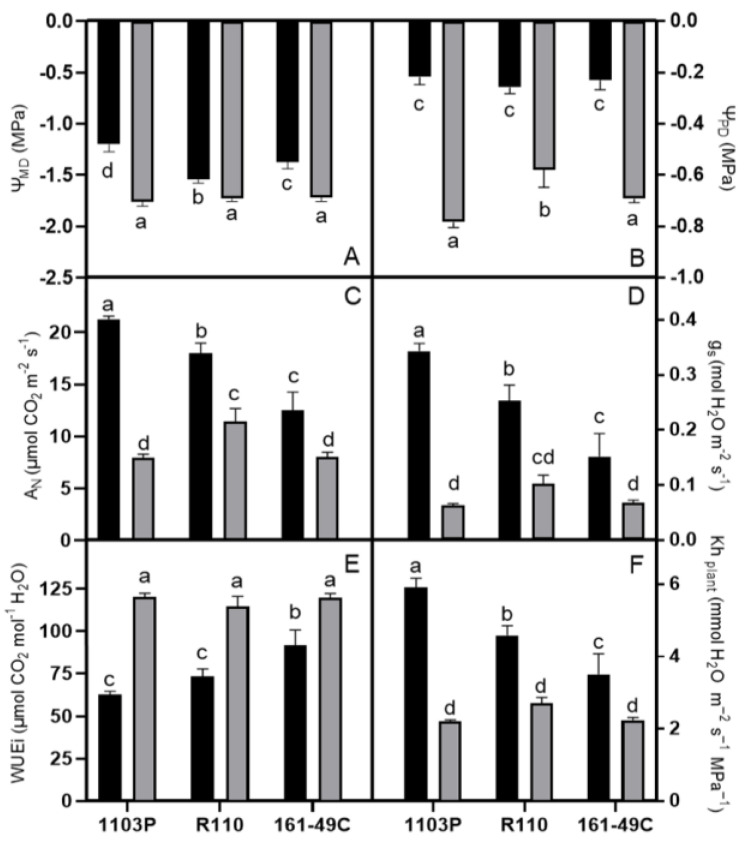
Effect of rootstocks and water stress (WS) on plant gas exchange and water status measurements. (**A**) Mid-day water potential (Ψ_MD_), (**B**) pre-dawn water potential (Ψ_PD_), (**C**) net photosynthesis (A_N_), (**D**) stomatal conductance (g_s_), (**E**) intrinsic water use efficiency efficiency (WUE_i_) and (**F**) whole plant hydraulic conductivity (Kh_plant_). Black and grey bars represent well water (WW) and WS treatments, respectively. Values represented are means ± standard error of each treatment and rootstock. Different letters denote statistically significant differences by one-way ANOVA with Duncan’s multiple comparison test (*p* < 0.05).

**Figure 2 plants-12-00718-f002:**
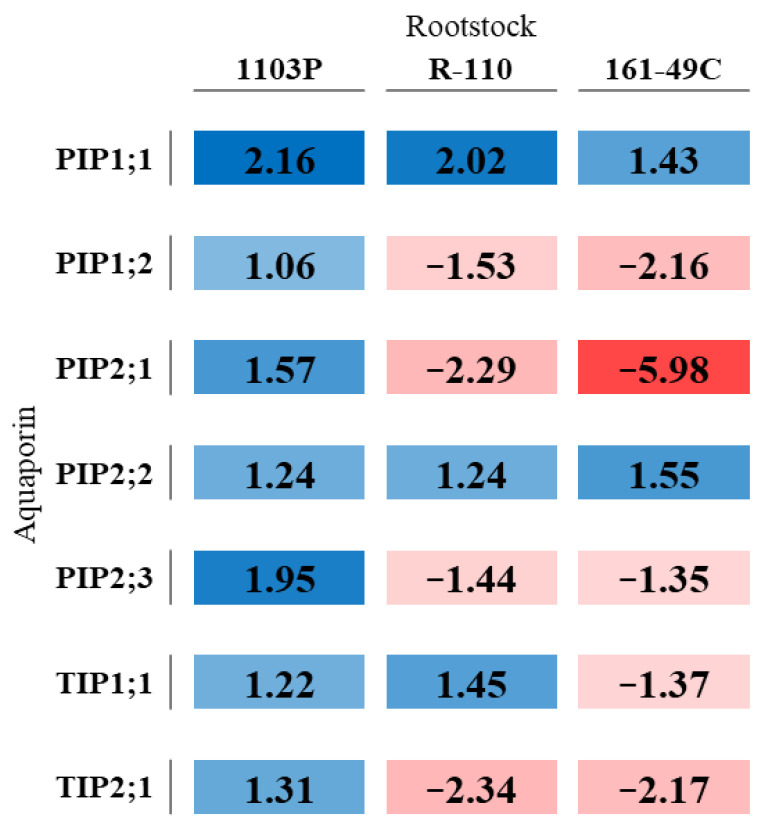
Aquaporin gene expression in each rootstock. Values represent the difference between well water (WW) and water stress (WS) expressions. A color gradient from blue (up-regulated) to red (down-regulated) shows the gene expression in each condition.

**Figure 3 plants-12-00718-f003:**
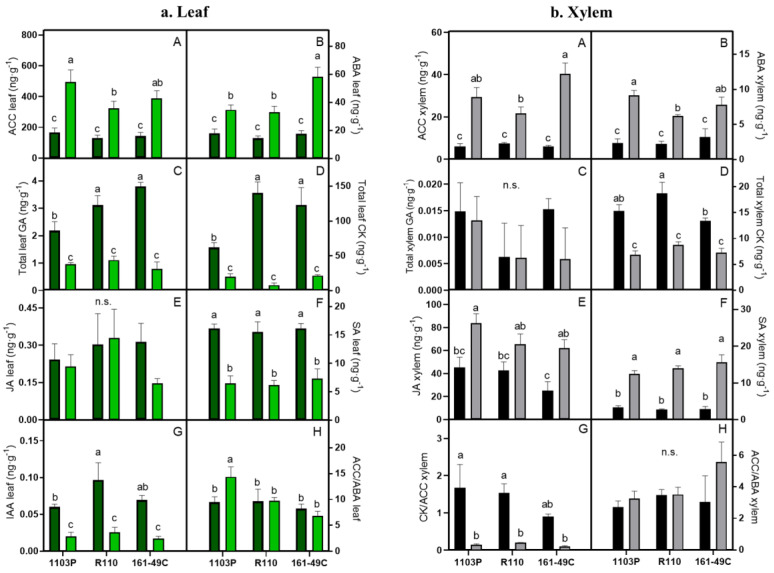
Leaf (**a**) and xylem (**b**) phytohormone concentrations under well water (WW) (dark green and black) and water stress (WS) (light green and grey) for the rootstocks studied. (A) Ethylene precursor 1-aminocyclopropane-1-carboxylic acid (ACC), (B) abscisic acid (ABA), (C) gibberellins (GA), (D) cytokinins (CK), (E) jasmonic acid (JA), (F) salicylic acid (SA), (G) indole-3-acetic acid (IAA) and (H) ACC/ABA ratio. Values represented are means ± SEM of each treatment and rootstocks. Different letters indicate statistically significant differences by one-way ANOVA with Duncan’s multiple comparison test (*p* < 0.05), while n.s. shows non-significant differences.

**Figure 4 plants-12-00718-f004:**
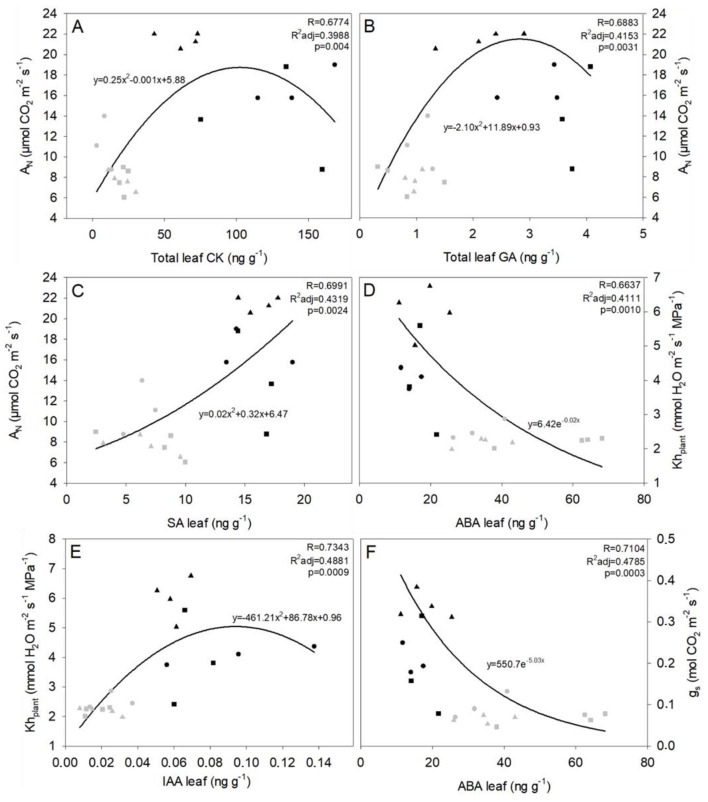
Correlation graphs between physiology and phytohormones. (**A**) Net photosynthesis (A_N_)- cytokinins (CK) regression, (**B**) net photosynthesis (A_N_)gibberellins (GA) regression, (**C**) net photosynthesis (A_N_)-salicylic acid (SA) regression, (**D**) whole plant hydraulic conductivity (Kh_plant_)-abscisic acid (ABA) regression, (**E**) whole plant hydraulic conductivity (Kh_plant_)-indole-3-acetic acid (IAA) regression and (**F**) stomatal conductance (g_s_)-abscisic acid (ABA) regression. The rootstocks (1103P (▲), R110 (●) and 161-49C (∎)) and treatments (WW: black; WS: grey) are represented on each graph. The green trend line shows that the relationship is between leaf parameters. Statistical parameters (R, R^2^ adjusted and *p*-value) also are indicated.

**Table 1 plants-12-00718-t001:** Mean of stomatal morphology and state measurements for treatments and rootstocks analyzed.

	Treatment		Rootstock			Significance		
	Well-Watered	Water-Stress	1103P	R110	161-49C	TR	RT	TR × RT
Density (stomatal number)	10.18 ± 1.37	9.96 ± 0.97	10.19 ± 0.86	10.36 ± 1.51	9.64 ± 1.07	ns	ns	ns
Width (µm)	16.49 ± 2.10	15.46 ± 1.31	15.67 ± 1.23	16.13 ± 2.12	16.15 ± 2.11	ns	ns	ns
Length (µm)	24.69 ± 1.91	22.53 ± 1.50	23.55 ± 2.05	23.60 ± 2.47	23.68 ± 1.62	**	ns	ns
Area (µm^2^)	336.23 ± 68.12	291.05 ± 37.89	307.36 ± 43.06	316.28 ± 77.28	317.86 ± 58.07	*	ns	ns

Values are means ± standard error of six vines per treatment. Significant differences for treatment (TR), rootstock (RT), and TR × RT were analyzed by two-way ANOVA (ns, not significant; *, *p* ≤ 0.05; **, *p* ≤ 0.01).

## Data Availability

The data analyzed during the current study are available from the corresponding author upon reasonable request.

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
