# Peer review of "The Rootstock Genotypes Determine Drought Tolerance by Regulating Aquaporin Expression at the Transcript Level and Phytohormone Balance"

_plants, 2023, doi:10.3390/plants12040718_

Round 1

Reviewer 1 Report

I just have several suggestions to authors.

quesiton1: line 521-522: “Measurement data were collected from one sun- 521 exposed leaf per plant, located at the base of a fertile green shoot.” Usually, authors need to point our which leave they measured. Since many gene expression levels are depended on the developmental stage of leaves. Especially, the expression of PIP genes is different in leaves of different leaves.

question2: Table 2 can be moved to the supplementary document, and the table 3 is not necessary at all. 

question 3: Table 1 showed that the RT doesn't affect the parameters of stomata significantly, however, the gas exchange parameters (FIgure 1), PIP expression levels (Figure 2) and phytohormone level (Fig. 3), all have quit great differences. I do believe the sampling methods is not fine designed, as I described in question 1. The difference may be caused by the methods.

Author Response

quesiton1: line 521-522: “Measurement data were collected from one sun- 521 exposed leaf per plant, located at the base of a fertile green shoot.” Usually, authors need to point our which leave they measured. Since many gene expression levels are depended on the developmental stage of leaves. Especially, the expression of PIP genes is different in leaves of different leaves. One leaf located in the main bell between nodes 8 and 9

Accordingly, we have added some more information in M&M (see line 547).

“Measurement data were collected from one sun-exposed leaf per plant, located in the main shoot between the nodes 8 and 9”

question2: Table 2 can be moved to the supplementary document, and the table 3 is not necessary at all. 

In this regard, we have removing Tables 2 and 3 from the manuscript and we have added Table 2 to supplementary material as Table S2.

question 3: Table 1 showed that the RT doesn't affect the parameters of stomata significantly, however, the gas exchange parameters (FIgure 1), PIP expression levels (Figure 2) and phytohormone level (Fig. 3), all have quit great differences. I do believe the sampling methods is not fine designed, as I described in question 1. The difference may be caused by the methods.

Gas exchange measurements and samplings (for aquaporin expression and phytohormones) were always carried out in the same plant and same kind of leaves (same position, same age, same exposure, etc.). Therefore, we believe that the experimental design is correct to compare all these parameters. It is true that the method of collecting stomata samples may affect the open or closed state of the stomata but not the stomatal morphology since the stomata are formed at the same time as the leaf and are not modified in the short term.

Reviewer 2 Report

In the revised manuscript authors tried to analyse impact of different rootstocks on the plant drought tolerance. They performed both physiological and molecular analyses. The research were focused on plant phytohormones and aquaporin genes expression.

In my opinion the presented study has a lot of shortcomings and needs broad improvement. Below are my detailed comments.

1. The structure of the manuscript is not straight enough.

2. Taking into consideration obtained results the title is nees to be changed. Authors concluded that physiological response of different rootstocks was similar under drought conditions.

3. The aim of the study nees to be more clear and more specific. Two assumptions included are almost similar.

4. Detail description of the plant material is missing. How rootstocks 1103P, R-110 and 161-49C diffgered between each other in drought tolerance? Did they represent differnt drought tolerance strategies like drought escape or avoidance. If the same scion was used for all rootstocks?

5. Details concerning water stress application are missing.

6. Why authors use the term semi-quantitative real-time PCR? The expression profiles of aquaporin genes should be presented in the form of a graph. Please, explain how the expression level was calculated. Why RNA was extracted from the leaves and not from the roots? Why only transcript level of aquaporin genes was determined? What about the protein level?

7. What the veraison is?

8. How Authors can explain the fact, that  the percentage of open stomata was not affected under drought?

9. The conclusions are too general.

Author Response

In the revised manuscript authors tried to analyse impact of different rootstocks on the plant drought tolerance. They performed both physiological and molecular analyses. The research were focused on plant phytohormones and aquaporin genes expression.

In my opinion the presented study has a lot of shortcomings and needs broad improvement. Below are my detailed comments.

  1. The structure of the manuscript is not straight enough.

Thank you for this comment, however after reading the article again, we really believe that the structure is the best way to follow the obtained results. It is the common structure used also by other authors when field experiments combine physiological parameters and laboratory analysis. The physiology of the plant is discussed first and then goes deeper into the molecular aspects. It is what we did in this work. Explaining the molecular results first and then the plant physiology could make it difficult to understand and follow.

  1. Taking into consideration obtained results the title is needs to be changed. Authors concluded that physiological response of different rootstocks was similar under drought conditions.

The title it is now changed accordingly. Thank you.

“The rootstock genotypes determine drought tolerance by regulating aquaporin expression, and phytohormone balance”.

  1. The aim of the study needs to be clearer and more specific. Two assumptions included are almost similar.

The aim has been rewritten to be more clear and specific (see lines 127-129)

  1. Detail description of the plant material is missing. How rootstocks 1103P, R-110 and 161-49C differed between each other in drought tolerance? Did they represent different drought tolerance strategies like drought escape or avoidance. If the same scion was used for all rootstocks?

Thank you for your comment. We consider that we explain rootstocks characteristics in introduction (see lines 66-70; 100-102), discussion (lines 290-296) and M&M (lines 518-521). However, we have added more information in the introduction about it.

  1. Details concerning water stress application are missing.

We explained that between lines 525 and 528.

“The two plots were differentially treated: plot 1 (well-water (WW)) had an irrigation dosage adjusted by a drip system up to 40% of the potential evapotranspiration (RTP); plot 2 (water stress (WS)) was not irrigated at all during the entire experimental period which lasted from April to May.”

  1. Why authors use the term semi-quantitative real-time PCR? The expression profiles of aquaporin genes should be presented in the form of a graph. Please, explain how the expression level was calculated. Why RNA was extracted from the leaves and not from the roots? Why only transcript level of aquaporin genes was determined? What about the protein level?

Firstly, we use the term semi-quatitative real-time PCR since we only measured up and down-regulation. Although we could quantify them, we did not quantify using a standard curve. Other authors like Dayer et al. 2020 and Gambetta et al 2012 used the term quantitative real-time PCR. However, Galmés et al. 2007 used the same term “semi-quatitative real-time PCR” to refer to it. That is why we thought that both were correct. In any case, we have changed in the manuscript (see line 577).

Secondly, the aquaporin expression is presented as a graph in the supplementary material (Figure S1). We showed the results as a heatmap because we thought that it was more visual and understandable. Besides, the method used to calculate the aquaporin expression was 2-∆∆Ct method. It is mentioned in the line 586. Anyway, we have added a brief method explanation (see lines 587-594).

Finally, we decided to analyze the leaves expression to be able to compare it with the plant physiology (measured in the leaves with LICOR). It is something innovative since most of the works focus on the roots expression. It would have been interesting to analyze the expression at the root level as well as the protein levels, but this was not possible for us. For future studies it would be a point to take into account.

  1. What the veraison is?

Veraison is a phenological stage characterized by the change of grape color from green to red and the accumulation of sugar in the grapes. It is the term used worldwide.

  1. How Authors can explain the fact, that the percentage of open stomata was not affected under drought?

We tried to calculated the percentage of open stomata. However, it was not possible for us. It is a mistake to have it in the results. Thanks for reporting this issue. We have solved it in the revised version of the manuscript (see line 176 and 178).

  1. The conclusions are too general.

Accordingly, we have include a brief and general conclusion since the magazine did not require the presence of this section. However, we have rewritten part of the conclusions to be more specific (see lines 487-506).

Round 2

Reviewer 1 Report

Here I really find the most critical problem of this manuscript it the way you design this research, in other words, the causal relationship among experiments you designed and made. 

In the last sentence of abstract, you declared "In conclusion, while rootstocks influenced stomatal control of water transpiration under WW conditions, implications on the molecular components of scion water transport and hormone concentration modify the scion drought response in a rootstock dependent manner." This conclusion has two problems as a scientific research conclusion: 1, what is the cause and what is the results? 2, Can the experimental data support this conclusion?

Here I cannot find clear answers in this manuscript. And in current plant physiology research, you need to provide genetic evidences or strong physiologic approaches to support your conclusion. For example, if you suggest the rootstocks influenced stomatal control of water transpiration under WW conditions. At least, you need to provide data from Vitis vinifera cv., without grafting, to avoid some physical problems might caused by grafting.

If you suggest hormone concentration can be a causal to the drought tolerance, you need to spray or inject the target hormone to the plants, and measure the physical parameters and gene expression levels. 

Therefore, I really suggest you to reconsider the experimental design and provide strong and logic data to support this manuscript. 

Author Response

Thank you for your comments, however, several aspects must be taken into consideration.

This experiment was carried out on well-established 30-year-old plants. All the studied plants within each rootstock were similar in shape and healthy. So, grafting problems were in this case avoided.

Moreover, it can be affirmed that the factors (rootstock and treatment) are the cause of the observed differences in scion physiology, aquaporin expression and phytohormone concentration. This is supported by a previous study published a few months ago (Pou et al. 2022) performed in the same experimental field, in which alterations at the scion physiological and nutritional levels were also different in a rootstock dependent-manner under irrigation conditions. Moreover, rootstocks have already been widely classified according to the drought resistance they provide (Lovisolo et al. 2016, Serra et al. 2014 and Alsina et al. 2011) but the mechanisms involved remain to be elucidated. As you mention, it would be ideal to test it on ungrafted Vitis vinifera. However, it is forbidden to test ungrafted Vitis vinifera in the field for sanitary reasons. Doing this experiment in pots would not allow us to compare the results with those obtained in the field since neither the age of the plants nor the environment, soil and space conditions, among others, would be the same and the results would not be comparable. Precisely the fact of having been able to carry out this study on 30-year-old plants allows us to obtain a more realistic vision of what is happening.

Moreover, other previous field studies in rootstocks such as those of Alsina et al. 2011, Koundras et al. 2008 and Romero et al. 2018 did not compare their results with ungrafted plants.

The rootstock affects the scion vigor, which affects the transpirational demand of the plant. In other words, the rootstock effect is not directly on aquaporins expression, but is a consequence of the effect on the scion. The same happens with many phytohormones that are synthesized in the aerial part of the plant. With those synthesized in the root and transported to the aerial part, a direct rootstock effect may be suggested. Therefore, the rootstock effect on the scion promotes changes at the physiological and molecular levels that provide different resistance to drought. To make it more clear and in line with what you mention, we have modified the text to clarify that rootstocks affect the scion differently, leading to alternative physiological and molecular responses.

Therefore, we believe that the experimental design is correct and that the results provided are reliable and strong. They allow us to suggest that both the differences in aquaporin expression and phytohormone concentrations are a clear evidence of differential response to water stress caused by rootstocks. Without them, the behavior observed in Tempranillo scion to water stress would be the same in all plants.

Reviewer 2 Report

Dear Authors,

the majority of my suggestions was taken into consideration. However, some aspects of the research still need improvement in my opinion.

1. I still do not like the title. I think it should be clarified that aquaporin genes expression was performed only at the transcript level. It is well-known that gene expression regulation is quite complex and transcript level do noit always correlates with protein level. It is also good to analyse protein activity to formulate proper conclusions. You have written that you are not able to measure aquaporin protein abundance. So, I suggest title change.

2. I think that root aquaporin expression should be performed. The reason is that you tried to analyse the impact of rootstock genotypes on drought tolerance. I suppose that rootstock and scion aquaporin genes can be regulated in the different manner.

3. I have comments concerning vertical axis description of the Figure S1. I suggest to use "normalized gene expression' description. Why in the figure legend authors wrote that dCt is presented?

4. Conclusions are still too general.

Author Response

  1. I still do not like the title. I think it should be clarified that aquaporin genes expression was performed only at the transcript level. It is well-known that gene expression regulation is quite complex and transcript level do noit always correlates with protein level. It is also good to analyse protein activity to formulate proper conclusions. You have written that you are not able to measure aquaporin protein abundance. So, I suggest title change.

Thank you, we have changed the title to specify that gene expression was at the transcript level. Now the title is:

“The rootstock genotypes determine drought tolerance by regulating aquaporin expression at the transcript level, and phytohormone balance”

  1. I think that root aquaporin expression should be performed. The reason is that you tried to analyse the impact of rootstock genotypes on drought tolerance. I suppose that rootstock and scion aquaporin genes can be regulated in the different manner.

We decided to analyze only the leaf expression of aquaporins at the leaf level based on results obtained previously for us in the same experimental field (Pou et al 2022). We observed that the rootstock had an effect on scion physiology and nutrition among others. Therefore, we wanted to analyze if it also affected aquaporin expression, adding the factor of drought.

Measuring the root expression would be very interesting but very difficult to do in a field experiment. The pot experiments that analyze it have no problem to pull up the plant and analyze the roots. However, it was not possible for us as we are dealing with a 30 year old commercial vineyard. Sampling without pull out the vines would also be complicated since it would not be possible to homogenize the sampling of all the plants. Therefore, it was not possible to analyze it for this study.

  1. I have comments concerning vertical axis description of the Figure S1. I suggest to use "normalized gene expression' description. Why in the figure legend authors wrote that dCt is presented?

Following your recommendation, we have changed the description of the vertical axis.

Regarding the legend, thanks to your comments, we have found an error. The value represented in the graphs corresponds to 2^ΔCt. It has been changed in the supplementary material. In this case it is not possible to represent the value of ΔΔCt since it relates irrigation and drought in the same value.

  1. Conclusions are still too general.

Thank you, although it is difficult for us to make it even more specific, we have tried to include more specific information and advice for future studies based on our results (please see lines 494516). We hope you find it interesting and specific enough.

Round 3

Reviewer 1 Report

OK, I understand the difficulties to do field experiments with a big population. It is somehow not a perfect research story, but it is acceptable.